# Cohort study of the characteristics and outcomes in patients with COVID-19 and in-hospital cardiac arrest

Astrid Holm ,[1] Matilda Jerkeman,[1] Pedram Sultanian ,[1] Peter Lundgren,[1,2] Annica Ravn-Fischer,[1] Johan Israelsson,[2,3] Jasna Giesecke,[4] Johan Herlitz,[5] Araz Rawshani [1]

¹Department of Molecular and Clinical Medicine, University of Gothenburg, Goteborg, Sweden
²Department of Internal Medicine, Kalmar County Hospital, Kalmar, Sweden
³Faculty of Health and Life Sciences, Linnaeus University, Kalmar, Sweden
⁴Clinicum-Centre for Clinical Skills, Interprofessional Education and Advanced Medical Simulation, Danderyd University Hospital, Stockholm, Sweden
⁵Centre for Prehospital Research, University of Borås, Borås, Sweden

**Correspondence to**
Dr Astrid Holm;
astrid.holm@gu.se

## ABSTRACT

**Objective** We studied characteristics, survival, causes of cardiac arrest, conditions preceding cardiac arrest, predictors of survival and trends in the prevalence of COVID-19 among in-hospital cardiac arrest (IHCA) cases.

**Design and setting** Registry-based observational study.

**Participants** We studied all cases (≥18 years of age) of IHCA receiving cardiopulmonary resuscitation in the Swedish Registry for Cardiopulmonary Resuscitation during 15 March 2020 to 31 December 2020. A total of 1613 patients were included and divided into the following groups: ongoing infection (COVID-19**+;** n=182), no infection (COVID-19**–;** n=1062) and unknown/not assessed (n=369).

**Main outcomes and measures** We studied monthly trends in proportions of COVID-19 associated IHCAs, causes of IHCA in relation to COVID-19 status, clinical conditions preceding the cardiac arrest and predictors of survival.

**Results** The rate of COVID-19+ patients suffering an IHCA increased to 23% during the first pandemic wave (April), then abated to 3% in July, and then increased to 19% during the second wave (December). Among COVID-19+ cases, 43% had respiratory insufficiency or infection as the underlying cause of the cardiac arrest, compared with 18% among COVID-19– cases. The most common clinical sign preceding cardiac arrest was hypoxia (57%) among COVID-19+ cases. OR for 30-day survival for COVID-19+ cases was 0.50 (95% CI 0.33 to 0.76), compared with COVID-19– cases.

**Conclusion** During pandemic peaks, up to one-fourth of all IHCAs are complicated by COVID-19, and these patients have halved chance of survival, with women displaying the worst outcomes.

## INTRODUCTION

The COVID-19 pandemic has, as of 6 November 2021, infected over 249 million individuals and lead to the death of over 5 million individuals.[1] COVID-19 is now the third-leading cause of death in Sweden.[2 3] Multiple studies have demonstrated that in-hospital cardiac arrest (IHCA) among patients with COVID-19 is associated with poor survival.[4–7] A recent study demonstrated that hypoxia

### Strengths and limitations of this study

► A major strength of our study is that it includes all in-hospital cardiac arrests (IHCAs) in Sweden which were reported to the Swedish Registry for Cardiopulmonary Resuscitation.
► The sample recorded in the Swedish Registry for Cardiopulmonary Resuscitation is unbiased since all hospitals participate in the registry and all hospitals report data on COVID-19 status.
► A limitation is that we do not know the severity of the COVID-19 infection, and we do not know if COVID-19 was the main reason for admission to hospital.
► Our study only includes IHCAs receiving cardiopulmonary resuscitation which leaves out all other patients with IHCA, for example, with a Do Not Attempt Resuscitation order.

was the main cause of cardiac arrest among 40% of patients with COVID-19 and IHCA.[6]

We have previously reported on COVID-19 and IHCA in the Swedish Registry for Cardiopulmonary Resuscitation (SRCR), showing a 2.3-fold increase in 30-day mortality among cases with COVID-19, compared with prepandemic cases. This was mainly driven by a nine-fold increase in mortality among women with COVID-19. At the time, no case of IHCA with COVID-19 had been discharged alive.[8] The current study expands our previous investigation, including more patients, longer follow-up and emphasises on the causes of cardiac arrest, predictors of survival, coexisting conditions and trends in the prevalence of COVID-19 among IHCA cases.

## METHODS
### Data sources

This study is a registry-based observational study with data obtained from the SRCR during the time period 15 March 2020 to 31 December 2020. The SRCR is a national quality registry and has included IHCA cases

since 2005. The data is collected by trained nurses who report patient data using a web-based protocol. The registry has previously been described in detail.[9] Vital status was obtained from the Swedish Population Registry and the last day of follow-up was 31 December 2020.

## Study population

The study population included all patients ≥18 years of age suffering IHCA and receiving cardiopulmonary resuscitation (CPR) throughout Sweden during the period 15March 2020 to 31 December 2020. We used 15 March as the start date of the pandemic as the Swedish Public Health Authority declared on 16 March 2020 that community spread had commenced.[3] On April, the SRCR started collecting data regarding COVID-19 status, and retrospectively identified 60 patients with COVID-19 who suffered IHCA during March (they were included in the study). Patients were divided into the following three groups: ongoing infection (COVID-19+; n=182), no infection (COVID-19−; n=1062) and unknown/not assessed (UNA; n=369). COVID-19 +was defined as patients registered with an ongoing COVID-19 infection, suspected ongoing infection or patients with a recent infection (n=29). The UNA group was included in the study in order to provide a complete picture of cases enrolled in the SRCR during the time period, and to evaluate whether missingness in COVID-19 status could entail selection bias.

## Variable definitions

In SRCR, a patient with cardiac arrest was defined as an unconscious patient with no or abnormal breathing, in whom resuscitation or defibrillation was attempted. IHCA was defined as cardiac arrest in patients admitted to the hospital.

With regard to previous coexisting conditions, heart failure was defined as any heart failure described before cardiac arrest. Kidney failure was defined as estimated glomerular filtration rate below $60\,mL/min/1.73\,m^2$, calculated using the highest creatinine before cardiac arrest with Chronic Kidney Disease Epidemiology Collaboration formula. The SRCR records data on the highest creatinine levels analysed up to 6months prior to CA. Diabetes was defined as any diabetes diagnosis, regardless of type. Cancer was defined as any previously known cancer. Acute myocardial infarction (MI) was defined as an MI within 72 hours of CA. Previous MI was defined as MI occurring earlier than 72 hours preceding the CA.

Regarding clinical conditions 1hour prior to CA, arrhythmia was defined as any arrhythmia, hypoxia was defined as an oxygen saturation below 90%, hypotension was defined as systolic blood pressure below 90 mm Hg, seizure was defined as any seizure with loss of consciousness, and heart failure was defined as any heart failure with pulmonary oedema or severe shortness of breath with rales.

Wards with monitoring included the coronary care unit (CCU), intensive care unit, operating room (OR),

emergency room (ER), high-dependency unit or the catheterisation laboratory.

## Statistical analyses

Patient characteristics are reported in means and medians, along with SD and IQRs, respectively. The Kaplan-Meier estimator was used for describing survival distributions; the log rank test was used to test for differences in survival. Trends in rates of COVID-19 were assessed on a monthly basis during the entire study basis.

Logistic regression was used to calculate ORs for 30-day survival. These models assessed the association between COVID-19 status and 30-day survival, adjusting for age, sex and initial rhythm (shockable or non-shockable). We performed subgroup analyses in relation to sex, age and coexisting conditions (heart failure, cancer, diabetes, kidney failure and MI). These subgroup analyses served to clarify whether the association between COVID-19 status and survival was modified by age, sex or coexisting conditions.

In order to obtain estimates of overall survival, we used Cox proportional hazards model with hours since CA as the time scale. The proportional hazards assumption was fulfilled for all variables.

We used the multiple imputation By chained equations) algorithm to impute missing values[10 11] (online supplemental figure 1). The imputed data set was used to calculate ORs for 30-day survival in the overall group, as well as in COVID+ and COVID− cases. These models included age, sex, initial rhythm, time to start of CPR, time of CA, previous MI, type of ward, heart failure, ECG monitoring, diabetes and acute MI.

Analyses were done in R (V.4.0.3, R Foundation for Statistical Computing) using RStudio.

## Patient and public involvement statement

No patients were involved.

## RESULTS

A total of 2227 patients were enrolled in the SRCR between 1 January 2020 and 31 December 2020. After excluding patients <18 years (n=68) and prepandemic cases (n=546), 1613 cases remained from 15 March 2020 to 31 December 2020 and constituted the final study population (online supplemental figure 2). There was a high rate of information on COVID-19 status during the study period among patients registered in the registry (online supplemental figure 3).

## Baseline characteristics

The overall mean age was 70.8 years, and the proportion of women was 37.6%. At the end of follow-up, 341 (32.7%) patients were alive. The mean age was similar in the three groups: 70.9 years in COVID+, 71.0 years in COVID− cases and 70.2 years in cases with UNA (online supplemental figure 4). The proportion of women was

**Table 1** Characteristics of 1613 patients with IHCA during the COVID-19 pandemic

| Variables | No infection COVID − | Ongoing infection COVID + | Unknown/NA UNA | SMD |
|---|---|---|---|---|
| n | 1062 | 182 | 369 | |
| Demographics: | | | | |
| Age—mean (SD) | 71.00 (13.32) | 70.93 (12.43) | 70.22 (13.60) | 0.039 |
| Woman—n (%) | 388 (36.6) | 68 (37.6) | 151 (41.0) | 0.061 |
| Location of cardiac arrest—n (%) | | | | 0.527 |
| Coronary care unit—n (%) | 155 (14.6) | 14 (7.7) | 50 (13.6) | |
| Intensive care unit—n (%) | 77 (7.3) | 25 (13.7) | 19 (5.1) | |
| Operational room—n (%) | 22 (2.1) | 0 (0.0) | 12 (3.3) | |
| Emergency room—n (%) | 139 (13.1) | 29 (15.9) | 65 (17.6) | |
| Outpatient lab, radiology—n (%) | 49 (4.6) | 7 (3.8) | 28 (7.6) | |
| Cathlab—n (%) | 98 (9.2) | 8 (4.4) | 60 (16.3) | |
| Intermediate care unit—n (%) | 25 (2.4) | 15 (8.2) | 10 (2.7) | |
| Regular ward—n (%) | 468 (44.1) | 82 (45.1) | 116 (31.4) | |
| Other—n (%) | 29 (2.7) | 2 (1.1) | 9 (2.4) | |
| Critical times—median (IQR): | | | | |
| Time to alert—median (IQR) | 1.00(1.00, 1.00) | 1.00(1.00, 1.00) | 1.00(1.00, 1.00) | 0.078 |
| Time to CPR—median (IQR) | 0.00(0.00, 1.00) | 0.00(0.00, 0.00) | 0.00(0.00, 1.00) | 0.109 |
| Time to defibrillation—median (IQR) | 2.00(1.00, 5.00) | 2.00(1.00, 4.75) | 1.00(1.00, 4.00) | 0.141 |
| Comorbidities—n (%): | | | | |
| MI, ongoing—n (%) | 178 (23.6) | 12 (12.0) | 37 (29.4) | 0.292 |
| MI, previous—n (%) | 163 (20.8) | 13 (11.7) | 26 (18.4) | 0.165 |
| Stroke, ongoing—n (%) | 30 (3.8) | 4 (3.7) | 4 (3.0) | 0.030 |
| Stroke, previous—n (%) | 82 (10.3) | 7 (6.1) | 15 (10.5) | 0.105 |
| Cancer, any—n (%) | 165 (20.9) | 20 (17.7) | 28 (20.6) | 0.054 |
| Diabetes—n (%) | 224 (27.9) | 36 (31.0) | 38 (27.0) | 0.060 |
| Heart failure—n (%) | 229 (29.7) | 36 (33.0) | 36 (27.9) | 0.074 |
| Ejection fraction (EF) (%)—mean (SD) | 46.14 (13.74) | 46.44 (11.86) | 44.94 (14.82) | 0.073 |
| EF <50% n (%) | 167 (46.0) | 26 (48.1) | 22 (46.8) | 0.029 |
| Kidney function category - n (%) | | | | 0.121 |
| eGFR <30 n (%) | 165 (21.6) | 22 (20.0) | 26 (20.0) | |
| eGFR 30–59 n (%) | 216 (28.3) | 32 (29.1) | 44 (33.8) | |
| eGFR 60–89 n (%) | 198 (25.9) | 25 (22.7) | 30 (23.1) | |
| eGFR ≥90 n (%) | 185 (24.2) | 31 (28.2) | 30 (23.1) | |
| No kidney failure (eGFR ≥60)—n (%) | 383 (50.1) | 56 (50.9) | 60 (46.2) | 0.063 |
| eGFR (mL/min/m$^2$)—mean (SD) | 66.89 (49.43) | 71.26 (58.96) | 63.78 (40.31) | 0.099 |
| Cause of arrest—n (%) | | | | 0.629 |
| Haemorrhage—n (%) | 34 (4.9) | 2 (2.0) | 10 (8.1) | |
| Myocardial infarction/ischaemia—n (%) | 181 (26.2) | 15 (14.9) | 41 (33.3) | |
| Other—n (%) | 213 (30.8) | 30 (29.7) | 41 (33.3) | |
| Primary arrhythmia—n (%) | 101 (14.6) | 8 (7.9) | 12 (9.8) | |
| Respiratory insufficiency—n (%) | 73 (10.5) | 24 (23.8) | 7 (5.7) | |
| Sepsis/infection—n (%) | 45 (6.5) | 19 (18.8) | 4 (3.3) | |
| Stroke/thromboembolism—n (%) | 45 (6.5) | 3 (3.0) | 8 (6.5) | |
| Early interventions—n (%): | | | | |

**Table 1**  Continued

| Variables | No infection COVID − | Ongoing infection COVID + | Unknown/NA UNA | SMD |
|---|---|---|---|---|
| Witnessed arrest—n (%) | 857 (80.9) | 140 (77.8) | 306 (85.0) | 0.124 |
| ECG monitoring—n (%) | 635 (60.5) | 89 (50.0) | 221 (62.1) | 0.163 |
| CPR before AGA - n (%) | 845 (91.0) | 146 (93.6) | 268 (88.2) | 0.127 |
| Defibrillated before AGA—n (%) | 159 (17.9) | 18 (11.9) | 53 (19.0) | 0.131 |
| Ventilated before AGA—n (%) | 503 (63.2) | 74 (54.8) | 175 (69.2) | 0.199 |
| Shockable rhythm—n (%) | 247 (24.9) | 29 (17.3) | 90 (27.0) | 0.158 |
| Defibrillated, any—n (%) | 323 (31.5) | 40 (22.7) | 111 (32.8) | 0.151 |
| Intubated—n (%) | 473 (47.0) | 100 (57.8) | 177 (53.8) | 0.145 |
| Epinephrine given—n (%) | 668 (65.6) | 125 (72.7) | 223 (66.4) | 0.102 |
| Antiarrhythmics—n (%) | 139 (14.1) | 17 (10.1) | 48 (15.4) | 0.107 |
| Mechanical compressions—n (%) | 109 (10.8) | 18 (10.4) | 66 (20.0) | 0.180 |
| Active temperature control—n (%) | 54 (11.3) | 5 (10.4) | 3 (4.4) | 0.173 |
| Status at rescue team arrival—n (%): | | | | |
| Consciousness—n (%) | 214 (23.1) | 18 (11.7) | 57 (19.3) | 0.204 |
| Breathing—n (%) | 288 (31.2) | 30 (19.5) | 84 (28.7) | 0.181 |
| Pulse—n (%) | 309 (33.8) | 36 (23.4) | 89 (30.4) | 0.154 |
| Follow-up data—n (%) | | | | |
| Angiography—n (%) | 115 (24.2) | 8 (16.7) | 15 (20.8) | 0.124 |
| PCI—n (%) | 87 (18.2) | 4 (8.3) | 16 (21.9) | 0.258 |
| Pacemaker implanted—n (%) | 80 (16.7) | 2 (4.2) | 4 (5.6) | 0.281 |
| ICD implanted—n (%) | 36 (7.5) | 1 (2.1) | 2 (2.8) | 0.172 |
| ROSC—n (%) | 520 (49.0) | 64 (35.2) | 142 (38.5) | 0.188 |
| Death at 30 days—n (%) | 666 (62.7) | 141 (77.5) | 237 (64.2) | 0.218 |
| Death overall—n (%) | 703 (66.2) | 141 (77.5) | 241 (65.3) | 0.181 |

SMD (difference between the means for the two groups divided by their mutual SD. Values below 0.1 (10%) are considered inconsequential (ie, no significant difference between the groups)).

AGA, alarm group arrival; CPR, cardiopulmonary resuscitation; eGFR, estimated glomerular filtration rate; ICD, implantable cardioverter-defibrillator; IHCA, in-hospital cardiac arrest; PCI, percutaneous coronary intervention; ROSC, return of spontaneous circulation; SMD, standardised mean difference; UNA, unknown/not assessed.

also similar; 37.6% in COVID+ and 36.6% in COVID− and 41.0% in UNA cases.

A regular ward was the most common place for cardiac arrest in all three groups; 45.1% of COVID+, 44.1% of COVID− and 31.4% of UNA cases occurred in regular wards (table 1). The ER was the second most common location for COVID+ cases (15.9%).

Regarding comorbidities, acute MI was observed in 12.0% of COVID+ and 23.6% of COVID− cases. Previous MI was observed in 11.7% of COVID+, 20.8% of COVID− and 11.7% of UNA cases. The prevalence of heart failure, cancer and diabetes was similar across all groups (table 1).

Fewer cases among COVID+ individuals had a shockable rhythm (17.3%), compared with COVID− (24.9%) cases. Likewise, fewer cases among COVID+ (22.7%) were defibrillated, compared with COVID− cases (31.5%). COVID+ cases were ventilated in 54.8% of cases before rescue team arrival, as compared with 63.2% in COVID− cases.

**Follow-up**

Return of spontaneous circulation after initial resuscitation, was less common in COVID+ cases, as compared with COVID− cases. Also, angiography, PCI, pacemaker and implantable cardioverter-defibrillator implantation postcardiac arrest were less common in COVID+ cases.

**Sex-specific characteristics**

Acute MI was observed in 21.2% of COVID +women and 7.6% of COVID +men. Previous MI was observed in 4.7% of COVID +women and 16.2% of COVID +men. The prevalence of previous stroke, renal failure, heart failure, cancer and diabetes was similar among men and women, as was location at the time of cardiac arrest. COVID + men were more likely to have a shockable rhythm (20.8%) compared with COVID +women (11.5%), and to be defibrillated (26.4% in men vs 16.9% in women) (online supplemental table 1).

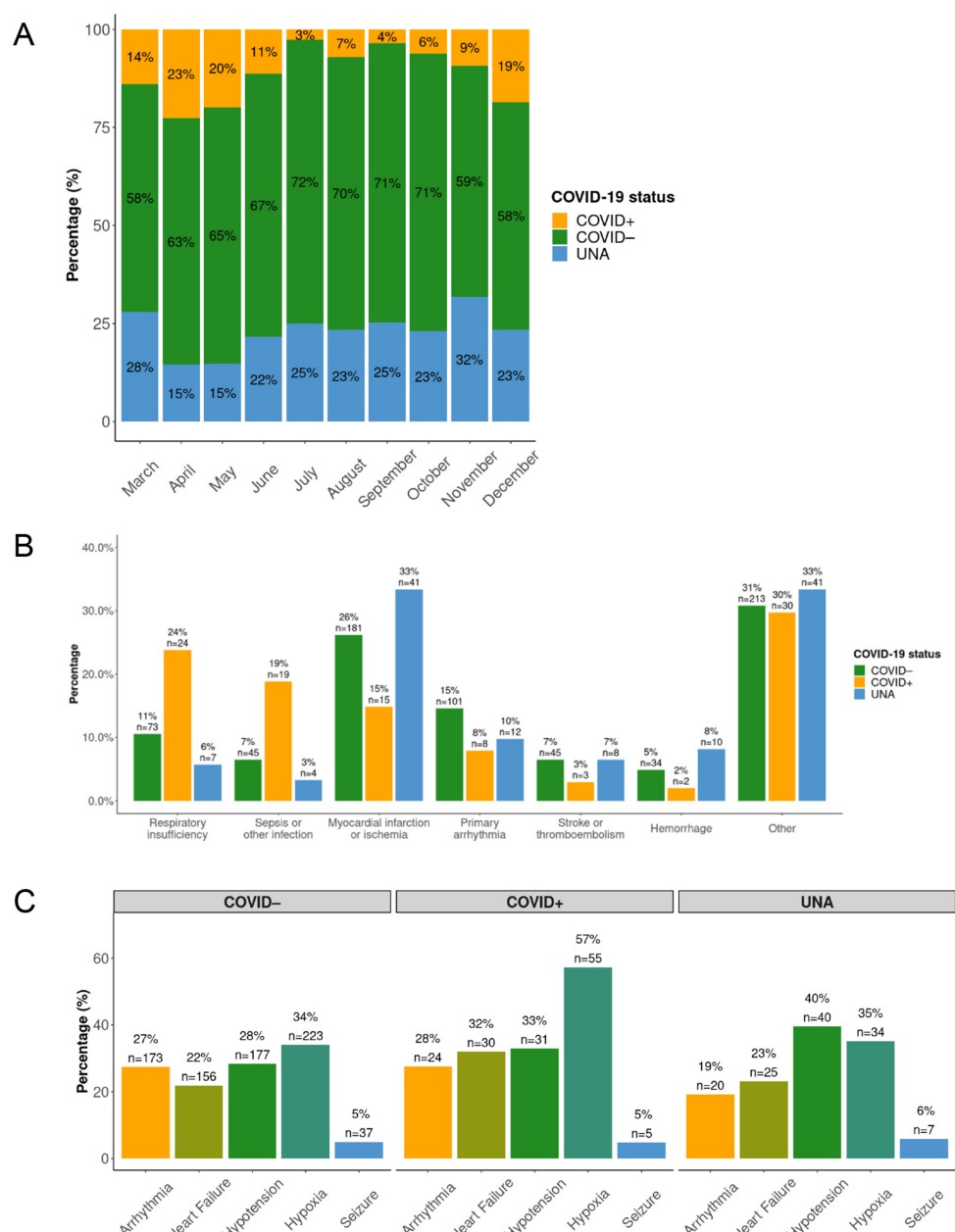

**Figure 1** Characteristics of in-hospital cardiac arrest (IHCA) according to COVID-19 status. (A) Monthly proportion of COVID-19 status among patients with IHCA, stratified on COVID-19 status. In March only cases after 15 March 2020 were included. (B) Aetiology of IHCA, stratified on COVID-19 status. The y-axis shows percentages for each aetiology in each group. (C) Clinical conditions 1 hour prior to IHCA, stratified on COVID-19 status. Only patients with data regarding the specific condition was included. UNA, unknown/not assessed.

### Monthly trends in COVID-19 associated IHCA

In March, April and May 14%, 23% and 20% of patients suffering IHCA were COVID+ (data from 16 March). The proportion of COVID + cases diminished rapidly during June to July. From September onwards, the COVID +cases increased again to reach 19% in December. In figure 1A, additional details regarding monthly variations are presented.

### Aetiology of IHCA

The most common cause of IHCA among COVID+cases was respiratory insufficiency (24%, n=24), and the second most common cause was sepsis or other infection (19%,

n=19). Respiratory insufficiency and sepsis/other infection were less common in the other groups (figure 1B), which instead displayed higher rates of acute MI.

### Clinical conditions one hour prior to IHCA

As evident in figure 1C, which describes the clinical conditions preceding (up to 60 min) the cardiac arrest, hypoxia was more common among COVID+ cases (57%), as compared with COVID– cases (34%).

### Survival analysis

The Kaplan-Meier plots (figure 2) show that COVID+ cases generally had a lower probability of survival compared

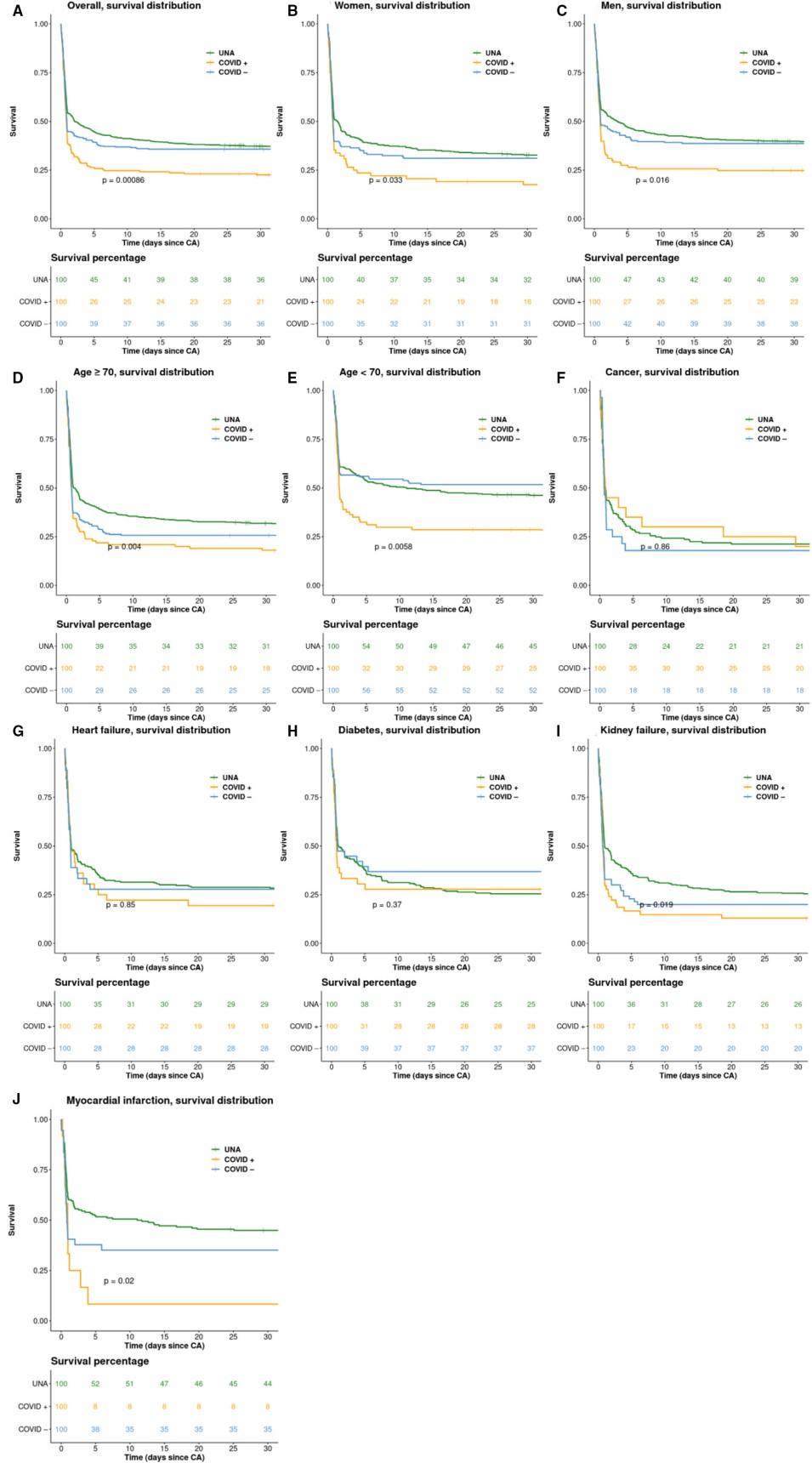

**Figure 2** Kaplan-Meier survival curves. Kaplan-Meier survival curves, separately for (A) overall, (B) women, (C) men, (D) age ≥70 year, (E) age <70 year, (F) cancer, (G) Heart failure, (H) diabetes, (I) kidney failure and (J) Myocardial infarction. p=log rank p value. The numbers under the graphs are showing the survival in percentages. regarding myocardial infarction acute MI is presented. CA, cardiac arrest; MI, myocardial infarction; UNA, unknown/not assessed.

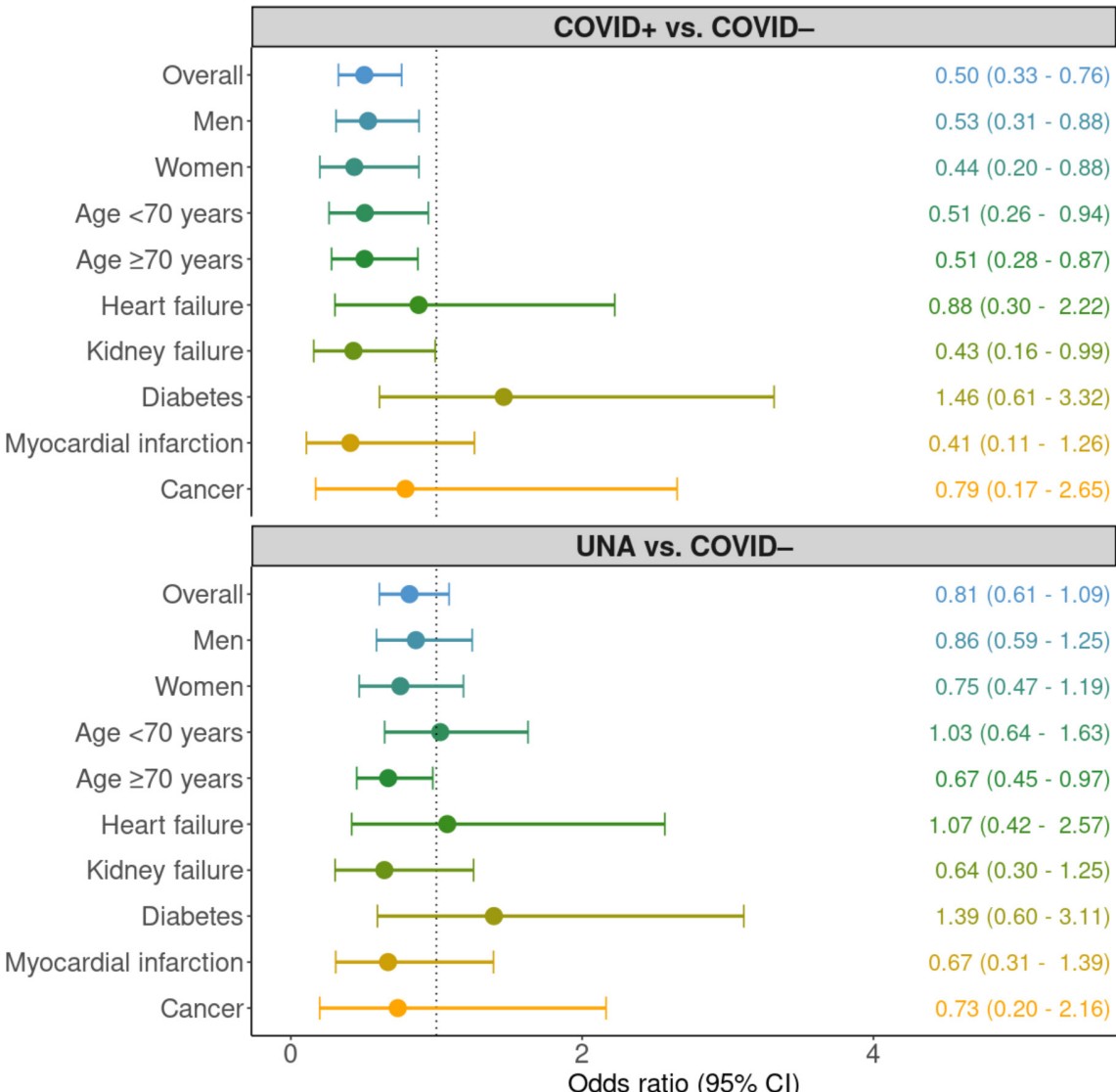

**Figure 3** OR for 30-day survival. Forest plot with the adjusted OR for 30-day survival among patients with ongoing infection versus no infection and unknown/NA vs no infection. Stratified on overall, men, women, age <70 years, age ≥70 years, heart failure, kidney failure, diabetes, myocardial infarction and cancer. Myocardial infarction (MI) was defined as acute or previous MI. UNA, unknown/not assessed.

with both COVID− and UNA cases. The overall 30-day survival (figure 2A) was 21% among COVID+, compared with 36% in COVID− cases (p=0.00086). The subgroup analysis of women (figure 2B) showed low survival rates in COVID+cases (16% 30-day survival). The subgroup analysis of men (figure 2C) showed low survival rates in COVID+cases (23% 30-day survival). The 30 days survival among COVID+ aged >70 years was 18% (figure 2D), as compared with 25% of COVID+ casesaged 70 years or younger (figure 2E). Survival curves for the subgroups of individuals with cancer, heart failure and diabetes, did not display any distinct patterns (figure 2F–2H), with all p>0.1. Patients with kidney failure had a 30 days survival of 13% among COVID+ cases (figure 2I). Patients with acute MI had a 30-day survival of 8% among COVID+ cases (figure 2J).

Cox adjusted survival curves are presented in online supplemental figure 5; COVID+ cases displayed the lowest probability of survival, whereas there was no material difference between COVID− and UNA cases.

### Odds ratios for 30-day survival
When adjusted for age, sex and initial rhythm the ORs for 30-day survival, comparing COVID+vs COVID−, were 0.50 (0.33–0.76) overall, 0.53 (0.31–0.88) for men and 0.44 (0.20–0.88) for women. In the subgroup of patients with heart failure, MI and cancer, we found no statistically significant associations, whereas in the subgroup of COVID+ patients with kidney failure, OR for 30-day survival was 0.43 (0.16–0.99), when compared with COVID− cases (figure 3).

### Predictors of survival
Regarding predictors of 30-day survival among COVID+, we note that CIs were generally wide. Lack of ECG

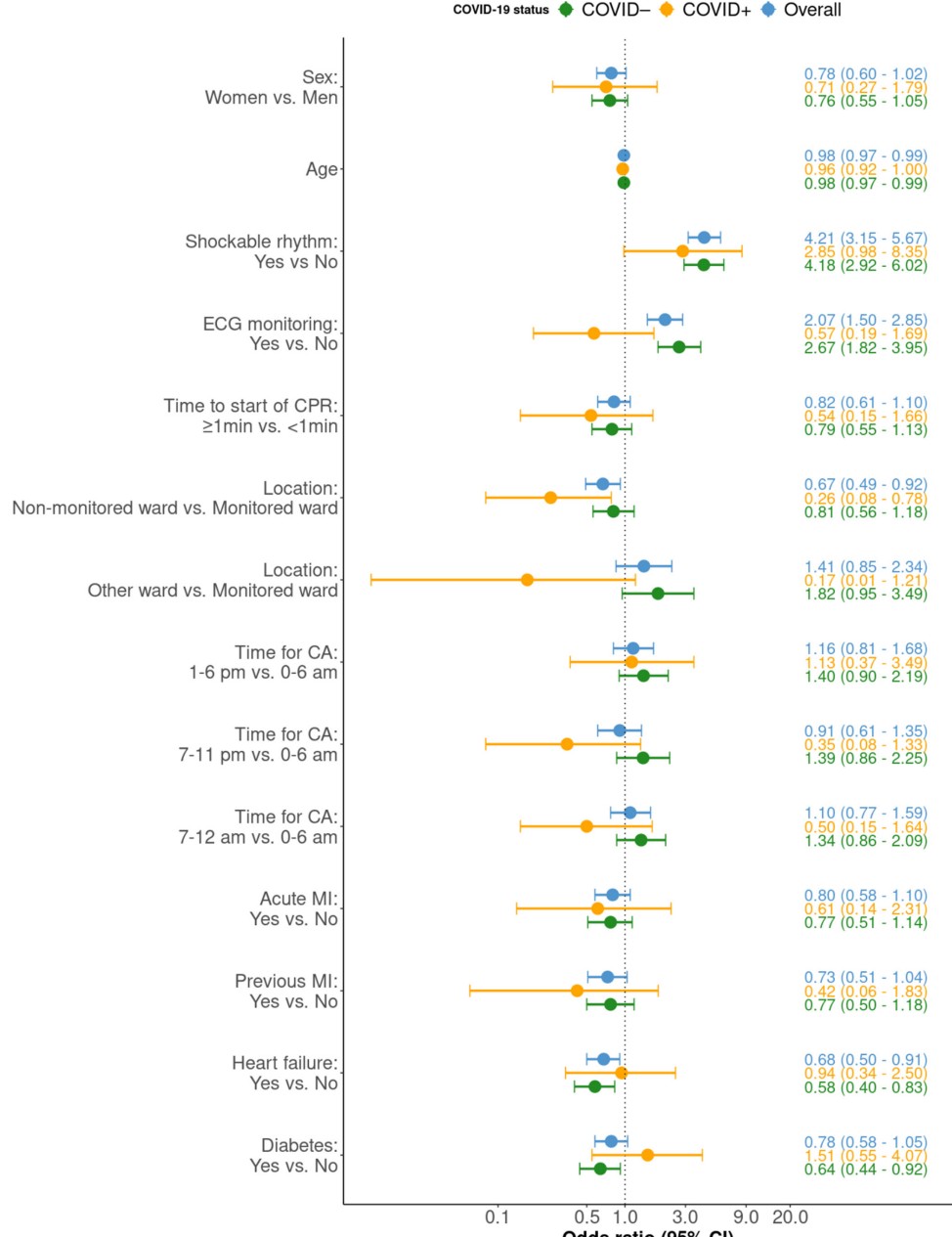

**Figure 4** OR for 30-day survival. Forest plot with OR for 30-day survival, stratified on the groups, no infection, ongoing infection and overall, all in different colours. The 95% CI is shown between the bars. X-axis has a logarithmic scale. CA, cardiac arrest; CPR, cardiopulmonary resuscitation; MI, myocardial infarction.

monitoring and delayed start of CPR showed point estimates below 1.0, although non-significant. OR for patients treated in non-monitored wards was 0.26 (95% CI 0.08 to 0.78) as compared with monitored wards (figure 4). No coexisting condition was associated with survival among COVID+ cases.

Among COVID-19– cases, the factors that were significantly associated with 30-day survival were shockable rhythm (OR 4.18 (95% CI 2.69 to 6.02)), ECG monitoring (2.67 (95% CI 1.82 to 3.95)), heart failure (OR 0.58 (95% CI 0.40 to 0.83)) and diabetes (OR 0.64 (95% CI 0.44 to 0.92); figure 4).

## DISCUSSION

This study elucidates characteristics and outcomes in patients with COVID-19 who develop IHCA. We show that the prevalence of COVID-19 among patients suffering an IHCA increased to approximately one in four cardiac arrests during the first pandemic wave, and one in five cardiac arrests during the second wave. In IHCA the probability of survival to 30 days is halved by the presence of COVID-19.

Regarding location of CA, we note that the most common location for COVID +patients was regular wards, which are not monitored. This is unfortunate since our

analyses showed that type of ward (monitored vs non-monitored) was significantly associated with survival, such that COVID + cases in non-monitored wards displayed 74% lower probability of survival as compared with COVID +cases in monitored wards. As compared with COVID– cases, cardiac arrest in the ER was more common in COVID +cases. The often rapid deterioration of cardiopulmonary function in patients with COVID-19 may be one of the explanations for this finding. Fewer COVID + cases were located in the CCU, which was an expected finding given that cardiac aetiology was less common among these patients.

We note that the most common cause of cardiac arrest in COVID+cases, as well as the most frequent clinical condition directly preceding the arrest, was respiratory. A total of 57% of cases displayed hypoxia before cardiac arrest. This may highlight an opportunity for improving outcomes; measures to prevent hypoxia and to correct it immediately may reduce the risk of cardiac arrest in patients with COVID-19. The high rate of respiratory aetiology was driven by men (online supplemental figures 6–7).

However, the fact that 43% of cases with COVID-19 did not have hypoxia prior to cardiac arrest suggests that other factors are important as well. Thromboembolism, MI, arrhythmias, etc may all contribute to the development of a cardiac arrest.[12]

A previous study from Wuhan showed that 87.5% of COVID+ cases with IHCA had a respiratory aetiology and a study from Southwest Georgia that 53% of the patients with IHCA and COVID-19 had ARDS.[5 7]

The survival rates were poor among COVID+patients with an overall 30-days survival of 21%, compared with 36% among COVID–. The survival rate was, however, not as low as in the study from Wuhan, in which 3% (151 patients studied) survived, or in the study from New York with 31 patients or in the study from Southwest Georgia with 63 patients with none surviving.[5 7 13] One reason for the poor survival could be the small number of patients found in shockable rhythm (17% vs 25% for COVID+ and COVID–, respectively) since patients with shockable rhythm have a more favourable outcome. After adjusting for sex, age and shockable rhythm the 30-day survival was still significantly worse among patients with an ongoing infection.

We demonstrate that COVID +women had halved chance of survival at 30 days, compared with COVID– women. We find it interesting that COVID +women had acute MI three times as often as men, despite the fact that men exhibited shockable rhythm—and were defibrillated—twice as often as women.

### Strengths and limitations

This study includes all IHCAs in Sweden which were reported to SRCR. The sample recorded in the SRCR is unbiased since all hospitals participate in the registry and all hospitals report data on COVID-19 status. However, we do not know the severity of the COVID-19 infection, and we do not know if COVID-19 was the main reason for admission to hospital. With regard to the classification of COVID-19 status, we have performed a misclassification analysis which demonstrated that ORs were not materially affected by misclassification bias. Missingness was prevalent with regards to cause of cardiac arrest, which is due to the difficulties determining this factor. However, we find no reason to believe that missingness differs across COVID-19 status categories, and it should, therefore, not bias our inferences. Our study only includes IHCAs receiving CPR. This leaves out all other patients with IHCA, for example, with a Do Not Attempt Resuscitation order.

Our regression models that included only COVID-19 cases should be interpreted with caution due to the large number of predictors in the model, with relatively few patients (resulting in wide CIs). Further studies are warranted, using a larger study population, and a longer follow-up especially regarding subgroup analyses, neurological outcomes and the quality of life for these patients.

### CONCLUSION

During pandemic peaks, up to one-fourth of all IHCAs are complicated by COVID-19, and these patients have halved chance of survival, with women displaying the worst outcomes.

**Contributors** AH and AR designed the study and were guarantors of the study. AH wrote the first draft of the manuscript, analysed all data and made initial interpretations of data. AR has been supervising. MJ, PS, PL, AR-F, JI, JG and JH revised the article critically for important intellectual content and approved the version of the article to be published.

**Funding** This work was supported by the Swedish Research Council (2019-02019) and the Swedish Heart and Lung Foundation (20 200 261).

**Competing interests** None declared.

**Patient consent for publication** Not applicable.

**Ethics approval** The study was approved by the Swedish Ethical Review Authority (ID 2020-02017). The data were anonymised before the authors accessed it for the purpose of the study.

**Provenance and peer review** Not commissioned; externally peer reviewed.

**Data availability statement** No data are available. No additional data available.

**ORCID iDs**
Astrid Holm http://orcid.org/0000-0003-4226-7494
Pedram Sultanian http://orcid.org/0000-0002-6941-6659
Araz Rawshani http://orcid.org/0000-0003-2066-3533

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
