## [Reviewer comments · BMJ Open]

ARTICLE DETAILS

TITLE (PROVISIONAL)	A Cohort Study of the Characteristics and Outcomes in Patients with COVID-19 and In-Hospital Cardiac Arrest
AUTHORS	Holm, Astrid; Jerkeman, Matilda; Sultanian, Pedram; Lundgren, Peter; Ravn-Fischer, Annica; Israelsson, Johan; Giesecke, Jasna; Herlitz, Johan; Rawshani, Araz

VERSION 1 – REVIEW

REVIEWER	Moskowitz, Ari Harvard University
REVIEW RETURNED	26-Jul-2021

GENERAL COMMENTS	Cardiac arrest among patients hospitalized with COVID-19 remains an important area of investigation. This study seeks to describe the epidemiology of cardiac arrest during COVID-19 in the Swedish national registry. Overall, I am excited to see ongoing research in this area. I do think there are a number of areas in which this manuscript could be improved— • The review of existing literature on this topic is incomplete and a number of important studies are not included. These include PMIDs—33093278, 33515638, 32283117, 32986117 among many others. The authors should perform another literature search and better place their manuscript within the contract of that literature.• The authors note that this study expands on their prior study where they demonstrated a substantially higher mortality rate for IHCA during the pandemic as compared to pre-pandemic. Many of the same patients are included in the present study and a very similar message is being presented. This overlap limits my enthusiasm for this study.• Its not entirely clear to me why the authors used survival analyses in this study. Was there a particular interest in time-to-event? The authors also use logistic regression to compare mortality between groups which seems more appropriate.• How did the authors decide on the studied subgroups? Given the relatively small number of COVID+ arrests, this is a large number of subgroups to explore.• There is a lot of focus throughout on sex-specific differences in survival. It is not clear to me that this was the focus a priori. This focus seems out of scope and might be better placed in a more targeted study.• There is a section of the Results focused on trends over time, although this analysis is not mentioned in the Methods section• I'm not sure I understand the 'causes of arrest' section. I find it unlikely that just 24% of COVID+ patients arrested due to respiratory insufficiency and there is likely substantial overlap with second most common cause 'sepsis or other infection.'• I am similarly surprised that just 57% of COVID patients who
---

	arrested in-hospital were hypoxemic. Are the authors able to validate this data to confirm it is not a coding error?  • The section on factors associated with survival is confusing to me—I'm not sure what the authors are trying to show here. For COVID- patients, many previous trials have extensively defined predictors of survival. For COVID+ patients, the cohort is very small for the number of potential predictors included. There is also no explanation for how the authors arrived at the potential predictors to include in the model. • It would be helpful to understand how the COVID- patients in this study compare to pre-pandemic patients. i.e. is the nature of arrest in COVID-19 – patients different if they arrest during or before the pandemic. This will also inform the comparison to COVID+ patients. • The Discussion is too long and largely just rehashes the results. It would be more helpful for the authors to better place their work in the broader context of COVID-19 research and explain more how they intend their results to be used by clinicians/investigators.
--	---

REVIEWER	Miles, Jeremy Jacobi Medical Center, Department of Medicine
REVIEW RETURNED	05-Aug-2021

GENERAL COMMENTS	Holm et al. in their manuscript entitled "A Cohort Study of the Characteristics and Outcomes in Patients with COVID-19 and In-Hospital Cardiac Arrest" elucidated the characteristics of patients diagnosed with COVID-19 in Sweden who experienced an in-hospital cardiac arrest and their outcomes as compared to patients without COVID-19 during a 10 month period in 2020. Overall, the manuscript was well-written and further elaborated on the increasing knowledge of the characteristics and outcomes of patients with COVID-19 who experience an IHCA in different parts of the world. Below are specific comments/questions:  1) Do all hospitals in Sweden have to report IHCA's to the Swedish Registry for Cardiopulmonary Resuscitation? A total of 183 IHCA's undergoing CPR in COVID positive patients throughout all of Sweden over a 10 month period during the peak of the pandemic seems a bit low. Is it known how many hospitalized patients died of COVID-19 in Sweden during this period? 2) The 369 patients who were in the 'unknown/not assessed' category, do we know any more about this group? Are these patients who experienced a cardiac arrest before a COVID test could be done? Or possibly just a missing data issue that wasn't available at the time the data was inputted into the registry? 3) There appear to be some errors in Table 1- it is written that 22 out of 182 COVID positive patients were discharged alive, which is 12%, not 19%. Also COVID negative patients discharged alive was 283 out of 1062 which is 27%, not 35%. Also the percentages of overall 30 day survival from Table 1 don't match up with the Kaplan Meier Curve percentages in Figure 2A 4) On page 15 it is written: "The most common cause of IHCA among COVID+ was respiratory insufficiency (24%, n=24). The second most common cause was sepsis or other infection (19%, n=19) among COVID+." There were a total of 182 IHCA among covid positive patients so these percentages don't correlate.
--

	5) Regarding clinical conditions prior to the cardiac arrest, 57% of patients with COVID-19 had hypoxia but only 24% of IHCA were considered secondary to respiratory insufficiency. How was etiology of the IHCA adjudicated? Is the process uniform among all hospitals in Sweden 6) 12% of COVID positive patients were found to have an acute myocardial infarction. How was AMI defined in the study cohort?
--	---

VERSION 1 – AUTHOR RESPONSE

Reviewer: 1

Dr. Ari Moskowitz, Harvard University

Comments to the Author:

Cardiac arrest among patients hospitalized with COVID-19 remains an important area of investigation. This study seeks to describe the epidemiology of cardiac arrest during COVID-19 in the Swedish national registry. Overall, I am excited to see ongoing research in this area. I do think there are a number of areas in which this manuscript could be improved—

1. The review of existing literature on this topic is incomplete and a number of important studies are not included. These include PMIDs—33093278, 33515638, 32283117, 32986117 among many others. The authors should perform another literature search and better place their manuscript within the context of that literature.

We thank the reviewer for this comment, which we agree with. We have therefore added additional references and comments (Introduction) with regards to this.

2. The authors note that this study expands on their prior study where they demonstrated a substantially higher mortality rate for IHCA during the pandemic as compared to pre-pandemic. Many of the same patients are included in the present study and a very similar message is being presented. This overlap limits my enthusiasm for this study.

This current study has a more than 50% larger study population of COVID+ patients (182 compared to 72 in the previous study); this study also covers a longer period of time, enabling us to present trends in COVID+ positive cardiac arrests and provide additional clinical details. All in all, we believe we had clinically significant data to this important research question.

3. It's not entirely clear to me why the authors used survival analyses in this study. Was there a particular interest in time-to-event? The authors also use logistic regression to compare mortality between groups which seems more appropriate.

We believe that some research questions are best answered by using logistic regression, notably 30-days survival. Other questions require us using survival time (Kaplan-Meier analyses). We did consider using Cox regression to maximize our ability to detect survival differences (which is hampered by the fact that the majority die within a day or two, resulting in many ties in the model). However, the Cox models resulted in results very similar to logistic regression, and we opted for the latter since it is in line with guidelines on reporting of cardiac arrest. Thus, we agree with the reviewer.

4. How did the authors decide on the studied subgroups? Given the relatively small number of COVID+ arrests, this is a large number of subgroups to explore.

Subgroups were defined according to the classifications available in the registry.

5. There is a lot of focus throughout on sex-specific differences in survival. It is not clear to me that this was the focus a priori. This focus seems out of scope and might be better placed in a more targeted study.

This is explained by the fact that in our previous study, we reported a 9-fold increased mortality in COVID+ cases and this was driven by mortality among women with COVID-19, prompting us to give it further consideration in this study.

6. There is a section of the Results focused on trends over time, although this analysis is not mentioned in the Methods section

We thank the reviewer for this comment. We have clarified in the Methods section. We have added: "Trends in rates of COVID-19 were assessed on a monthly basis during the entire study period".

7. I'm not sure I understand the 'causes of arrest' section. I find it unlikely that just 24% of COVID+ patients arrested due to respiratory insufficiency and there is likely substantial overlap with second most common cause 'sepsis or other infection.'

In the registry it is only possible to record the primary cause of cardiac arrest. The attending clinician is responsible for determining what the main cause of the arrest was. Thus, the reviewer is correct that the patients probably had multiple causes of cardiac arrest and this is regrettably not elucidated by our study.

8. I am similarly surprised that just 57% of COVID patients who arrested in-hospital were hypoxemic. Are the authors able to validate this data to confirm it is not a coding error?

This is certainly a valid objection. Hypoxemia was defined as saturation <90%. Fortunately, we can confirm that this figure is correct and it may be explained by the fact that COVID infections may cause serious events by multiple avenues, including thrombosis, ARDS, sepsis, arrhythmias, etc.

9. The section on factors associated with survival is confusing to me—I'm not sure what the authors are trying to show here. For COVID- patients, many previous trials have extensively defined predictors of survival. For COVID+ patients, the cohort is very small for the number of potential predictors included. There is also no explanation for how the authors arrived at the potential predictors to include in the model.

We thank the reviewer for this comment. We have now added the following to the Methods section: "For these models we selected variables that are established predictors of survival after IHCA and suspected confounders." This analysis is exploratory in nature, knowing that we have rather few cases with COVID, yet an important analysis to attempt given the gaps in knowledge and severity of the disease.

10. It would be helpful to understand how the COVID- patients in this study compare to pre-pandemic patients. i.e. is the nature of arrest in COVID-19 – patients different if they arrest during or before the pandemic. This will also inform the comparison to COVID+ patients.

We thank the reviewer for this comment. We will attempt to perform these analyses in a subsequent investigation, as we believe it will require a number of in-depth analyses.

11. The Discussion is too long and largely just rehashes the results. It would be more helpful for the authors to better place their work in the broader context of COVID-19 research and explain more how they intend their results to be used by clinicians/investigators.

We thank the reviewer for this comment. We have shortened the Discussion and put the results into a broader context, as asked for.

Reviewer: 2

Dr. Jeremy Miles, Jacobi Medical Center

Comments to the Author:

Holm et al. in their manuscript entitled "A Cohort Study of the Characteristics and Outcomes in Patients with COVID-19 and In-Hospital Cardiac Arrest" elucidated the characteristics of patients diagnosed with COVID-19 in Sweden who experienced an in-hospital cardiac arrest and their outcomes as compared to patients without COVID-19 during a 10 month period in 2020.

Overall, the manuscript was well-written and further elaborated on the increasing knowledge of the characteristics and outcomes of patients with COVID-19 who experience an IHCA in different parts of the world. Below are specific comments/questions:

1) Do all hospitals in Sweden have to report IHCAs to the Swedish Registry for Cardiopulmonary Resuscitation? A total of 183 IHCAs undergoing CPR in COVID positive patients throughout all of Sweden over a 10 month period during the peak of the pandemic seems a bit low. Is it known how many hospitalized patients died of COVID-19 in Sweden during this period?

All (100%) of hospitals in Sweden report their cases to the registry. We do retrospective assessments of coverage and have around 95% level of ascertainment regarding IHCA. However, the registry only includes cases in whom CPR was attempted. We have performed validations over the years to assess how many patients with IHCA who have a DNAR decision (do not attempt resuscitation) and concluded that roughly 90% of in-hospital IHCAs are DNAR. It does not lie in the interest of the registry to study those cases, as the registry only studies management and outcomes of hearts that should be saved. Nevertheless, the pandemic has been difficult for the entire health care system, including those reporting data to the registry and in some cases reporting is done retrospectively, such that additional cases may be reported by the end of the year. This does not, however, result any systematic bias with regards to our results, but it does reduce power.

2) The 369 patients who were in the 'unknown/not assessed' category, do we know any more about this group? Are these patients who experienced a cardiac arrest before a COVID test could be done? Or possibly just a missing data issue that wasn't available at the time the data was inputted into the registry?

The reviewer is correct. This are patients with COVID status unknown at the time of reporting. We decide to show these patients as a separate group in order to provide full data transparency.

3) There appear to be some errors in Table 1- it is written that 22 out of 182 COVID positive patients were discharged alive, which is 12%, not 19%. Also COVID negative patients discharged alive was 283 out of 1062 which is 27%, not 35%. Also the percentages of overall 30 day survival from Table 1 dont match up with the Kaplan Meier Curve percentages in Figure 2A

We thank the reviewer for noticing this. The variable “Discharged alive” is poor due to missingness (as evident in the previous version). We have removed it completely from the study. The mortality variables are based on data from the vital status registry, which is reliable and used in virtually all Swedish outcome studies. We have re-checked survival data in the K-M plots and 30-days survival in Table 1 and they are concordant.

4) On page 15 it is written: "The most common cause of IHCA among COVID+ was respiratory insufficiency (24%, n=24). The second most common cause was sepsis or other infection (19%, n=19) among COVID+." There were a total of 182 IHCA among covid positive patients so these percentages don't correlate.

We thank the reviewer for this comment. This is a reflection of the difficulty determining the primary cause of the arrest. Lack of certainty results in missing data, for this variable. We have clarified this in the Limitations now, by adding the following sentence “Missingness was prevalence for cause of cardiac arrest, which is due to the difficulties determining this. However, we find no reason to believe that missingness rates differ across our study groups”.

5) Regarding clinical conditions prior to the cardiac arrest, 57% of patients with COVID-19 had hypoxia but only 24% of IHCAs were considered secondary to respiratory insufficiency. How was etiology of the IHCA adjudicated? Is the process uniform among all hospitals in Sweden

The data is collected by trained nurses who report patient data using a web-based protocol. The protocol is the same in Sweden. Reporting is done in three sequences and in the final sequence, the trained nurse re-assesses the cause of cardiac arrest. In case of uncertainty, a physician or the registry itself can be consulted. Yet, there can be significant missingness due to the inherent difficulty establishing cause of cardiac arrest.

6) 12% of COVID positive patients were found to have an acute myocardial infarction. How was AMI defined in the study cohort?

Acute myocardial infarction (MI) was defined as an MI within 72 hours of CA. The definition of acute MI is based on guidelines issued by the European Society for Cardiology (ESC).

VERSION 2 – REVIEW

REVIEWER	Moskowitz, Ari Harvard University
REVIEW RETURNED	15-Oct-2021

GENERAL COMMENTS	Thank you for your responses and edits to the manuscript. Overall the manuscript is improved, however I continue to have some reservations on the overall impact of the findings above and beyond what has already been published in the literature regarding
--

	generally poor outcomes after COVID-19+ IHCA. Specific comments to follow:  1. I would de-emphasize or fully remove the focus on cause of arrest. As the authors note, there were many potential categories to select from and only one could be selected. To say that many COVID-19+ IHCA were not due to respiratory insufficiency may be misleading. 2. I'm not sure what to make of the UNA category. More patients were UNA than were COVID+. I would consider simply excluding that group from the analyses. 3. I continue to feel that there are an overly large number of subgroups analyzed and there is no description in the Methods explaining why these subgroups were selected. 4. The updated line in the Conclusion "The Pandemic has changed the whole world and the halved chance of survival displays just a little part of how it has affected us all" is overly colloquial and not supported by the data presented.
--	---

VERSION 2 – AUTHOR RESPONSE

Reviewer: 1

Dr. Ari Moskowitz, Harvard University

Comments to the Author:

Thank you for your responses and edits to the manuscript.

Reply: We are grateful to Dr Moskowitz for his continued efforts in propelling our manuscript forward with constructive critique.

1. I would de-emphasize or fully remove the focus on cause of arrest. As the authors note, there were many potential categories to select from and only one could be selected. To say that many COVID-19+ IHCA were not due to respiratory insufficiency may be misleading.

REPLY: We thank the reviewer for this comment. Please refer to the third paragraph in the Discussion, where we clarify that it is certainly the respiratory causes driving cardiac arrests in COVID+ patients. We write:

“We note that the most common cause of cardiac arrest in COVID+ cases, as well as the most frequent clinical condition directly preceding the arrest, is respiratory. The high rate of respiratory etiology was driven by men (Supplementary Figure 6-7). A total of 57% of cases displayed hypoxia before cardiac arrest. This may highlight an opportunity for improving outcomes; measures to prevent hypoxia and to correct it immediately may reduce the risk of cardiac arrest in patients with COVID-19.”

2. I'm not sure what to make of the UNA category. More patients were UNA than were COVID+. I would consider simply excluding that group from the analyses.

REPLY: We decided to show these patients as a separate group in order to provide full data transparency. This was discussed several times in our group and all authors agreed that this group

should be included in order to declare the risk of selection bias in testing/assessing COVID status. We added the following sentence to Methods > Study Population:

“The UNA group was included in the study in order to provide a complete picture of cases enrolled in the SRCR during the time period, and to evaluate whether missingness in COVID-19 status could entail selection bias.”

3. I continue to feel that there are an overly large number of subgroups analyzed and there is no description in the Methods explaining why these subgroups were selected.

REPLY: We have now added a brief justification to under statistical analyses, stating the following:

“For survival analyses, we performed subgroup analyses in relation to sex, age and coexisting conditions (heart failure, cancer, diabetes kidney failure and myocardial infarction). These subgroup analyses served to clarify whether the association between COVID status and survival was modified by age, sex or coexisting conditions.”

4. The updated line in the Conclusion "The Pandemic has changed the whole world and the halved chance of survival displays just a little part of how it has affected us all" is overly colloquial and not supported by the data presented.

REPLY: We agree and have removed that sentence completely. The conclusion now reads:

“During pandemic peaks, up to one fourth of all IHCAAs are complicated by COVID-19, and these patients have halved chance of survival, with women displaying the worst outcomes.”.